# MatchNAS: Optimizing Edge AI in Sparse-Label Data Contexts via Automating Deep Neural Network Porting for Mobile Deployment

Submission Id: 1301

## ABSTRACT

Recent years have seen the explosion of edge intelligence with powerful deep learning models. As 5G technology becomes more widespread, it has opened up new possibilities for edge intelligence, where the cloud-edge scheme has emerged to overcome the limited computational capabilities of edge devices. Deep-learning models can be trained on powerful cloud servers and then ported to smart edge devices after model lightweight. However, porting models to match a variety of edge platforms with real-world data, especially in sparse-label data contexts, is a labour-intensive and resource-costing task. In this paper, we present MatchNAS, a neural network porting scheme, to automate network porting for mobile platforms in label-scarce contexts. Specifically, we employ neural architecture search schemes to reduce human effort in network fine-tuning and semi-supervised learning techniques to overcome the challenge of lacking labelled data. MatchNAS can serve as an intermediary that helps bridge the gap between cloud AI and edge AI, facilitating both porting efficiency and network performance.

## KEYWORDS

Edge AI, mobile intelligence, deep neural network, AutoML

### ACM Reference Format:

Anonymous Author(s). 2024. MatchNAS: Optimizing Edge AI in Sparse-Label Data Contexts via Automating Deep Neural Network Porting for Mobile Deployment. In *Proceedings of the ACM Web Conference 2024 (WWW '24), May 13–17, 2024, Singapore.* ACM, New York, NY, USA, 11 pages. https://doi.org/XXXXXXX.XXXXXXX

## 1 INTRODUCTION

Recent years have seen the popularity of artificial intelligence (AI) and deep learning (DL) not only in server-based platforms but also in edge devices. AI and DL have been applied in a wide spectrum of edge applications such as Internet-of-Things (IoT), Web-of-Things (WoT) and mobile intelligence, powering edge devices to become "smart", such as real-time image analytics [15], natural language recognition [10], health monitoring [42], personal recommendation systems [27], etc.

*WWW '24, May 13-17, 2024, Singapore*
© 2024 Association for Computing Machinery.
ACM ISBN 978-1-4503-XXXX-X/18/06…$15.00
https://doi.org/XXXXXXX.XXXXXXX

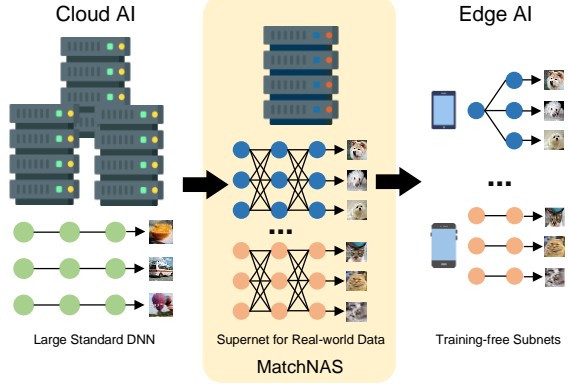

**Figure 1: MatchNAS bridges the Cloud AI and Edge AI.**

Current DL-based functionalities and applications can be attributed to the rapid development of Deep Neural Networks (DNNs). DNNs are typically a data-hungry paradigm that necessitates large quantities of labelled data for model training [26] to achieve better performance. To digest and absorb the "knowledge" hidden in huge volumes of data, DNNs have become "deeper" with more network parameters and computing complexity. However, edge intelligence cannot benefit from large DNNs directly because they have limited computing resources and low computational capability.

In this regard, cloud-edge DNN porting gained popularity [6, 8, 11, 12] for edge AI with the proliferation of 5G mobile networks [14, 23]. This porting scheme leverages powerful cloud-based servers to train large DNNs with amounts of data and deploy them to edge devices after architectural compressing and fine-tuning with real-world data [39]. Although promising, this DNN porting scheme has two major bottlenecks. Firstly, tailoring networks for varying edge platforms is a labour-costing task requiring great human efforts and computing resources as the number of platforms increases [5, 34]. Secondly, real-world data tends to be label-scarce [4, 33] since labelling data is labour-intensive and expertise-requiring. Lightweight networks with fewer parameters struggle to handle large quantities of unlabelled data, increasing network fine-tuning difficulty. As mobile computing becomes increasingly important, a large bunch of model light-weighting techniques and training strategies have been proposed to support mobile intelligence.

Recent researchers studied Neural Architecture Search (NAS) to automate mobile porting. Zero-shot NAS [7, 21, 25] designs specific metrics for model performance evaluation, which can effectively obtain optimum network architectures for different platforms, avoiding manual network designing. One-shot NAS [5, 13, 30] jointly trains a huge network family with similar architecture to avoid

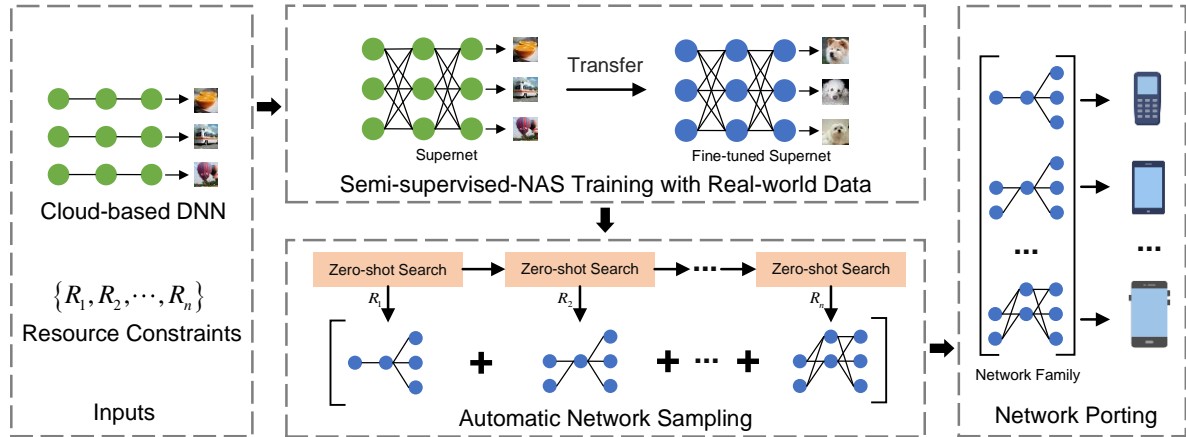

**Figure 2: A workflow for MatchNAS. Given a pre-trained cloud-based DNN and a set of resource constraints, MatchNAS first transforms the DNN to a supernet and inherits its network weights. Then, MatchNAS conducts a semi-supervised-NAS training, which is a combination of semi-supervised learning and one-shot NAS, to transfer the supernet to a label-scarce dataset. After training, MatchNAS leverages the zero-shot NAS techniques to efficiently sample high-quality subnets from the supernet according to the resource constraints without further training and build a network family for efficient network mobile porting.**

repeated training for each platform. These NAS techniques accelerate the porting progress for diverse edge platforms. However, NAS training is a data-hungry task, acquiring large amounts of labelled data, which will result in poor performance in label-scarce contexts.

As for the second bottleneck, SSL algorithms [3, 4, 33, 35, 38] make effective use of both labelled and unlabelled data. For example, pseudo-labelling [20] produces artificial labels based on the model's prediction and trains the model to predict the artificial labels when feeding unlabelled data. However, the performance of an SSL-based model is highly correlated to the quality of artificial labels. Constrained by limited network parameters, lightweight models may have difficulty producing high-quality artificial labels. In addition, current mainstream research on SSL does not take into account mobile porting or NAS.

In this work, we focus on improving the DNN porting for mobile deployment. To overcome the above-mentioned two bottlenecks in mobile porting, we propose an automatic DNNs porting algorithm to bridge the cloud AI and the edge AI, namely MatchNAS, by fully utilising the techniques in NAS and SSL. As shown in Figure 2, given a pre-trained cloud-based network, we first transform it to a "supernet" (i.e., a set of sub-networks with a similar architecture). Then, we train this supernet with semi-supervised techniques to support label-scarce datasets. Specifically, during supernet training, MatchNAS leverages the largest subnet in the family to produce high-quality artificial labels for other smaller candidates. Through this scheme, the "knowledge" from the largest network can be distilled to lighter networks for performance improvement. After training, we leverage the zero-shot NAS technique to directly obtain optimal sub-networks for network porting without repeated training. To the best of our knowledge, this paper is the first to accelerate and improve mobile porting within label-limited contexts. Our main contributions are as follows:

(1) We propose MatchNAS, a semi-supervised-NAS algorithm to optimize edge AI by automating mobile DNN porting.

Through MatchNAS, we greatly reduce the training cost and improve network performance for mobile deployment.

(2) In this paper, we evaluate MatchNAS on four image classification datasets with limited labelled data, including Cifar-10 [19], Cifar-100 [19], Cub-200 [36] and Stanford-Car [18]. Compared to the state-of-the-art NAS [5, 13] and SSL [33] methods, MatchNAS achieve a higher network performance, for example, a maximum 20% Top-1 accuracy improvement on Cifar100 (4000 labelled examples) with 15M Floating Point Operations (FLOPs).

(3) We deploy MatchNAS's networks to a couple of popular smartphones, and MatchNAS show a better latency-accuracy trade-off compared to the SOTA methods.

## 2 RELATED WORK

Considering the resource constraints, many in-the-wild DL applications for edge AI are very lightweight [39]. There are two issues for porting lightweight DNNs. Firstly, the human effort in manual network compressing and training cost in network optimization increases significantly as the number of target platforms and deployment scenarios increases. Secondly, lightweight networks have trouble encountering sparse-label data contexts of limited network parameters. In this section, we will introduce neural architecture search and semi-supervised learning for their advantages and disadvantages in mitigating these two issues.

### 2.1 Neural Architecture Search

Neural Architecture Search (NAS) [2, 43] has gained widespread attention for automating network design with less manual intervention. The general idea behind NAS is to explore network architecture from a space of different architectural choices, such as the number of layers and operation types. This enables the creation of resource-insensitive models for mobile deployment [5, 30, 34]. Early NAS [29, 32, 43, 44] suffers from prohibitive resource consumption

of training and evaluating every candidate networks, while recent one-shot NAS and zero-shot NAS alleviate this burden by supernet training scheme and architectural scoring scheme, respectively.

Let $\mathcal{A}$ be a search space containing a set of candidate networks $\{\alpha_i\}$ with the same functionality but different architectural configurations, such as the number of layers, the size of convolution kernels, the number of channels, etc. Let $N$ be the total number of candidate architecture in $\mathcal{A}$. Let $D^{\text{trn}} = \{(x, y)\}$ be the labelled training datasets where $x$ is a batch of training inputs and $y$ is corresponding labels, and $D^{\text{val}}$ is the validation datasets. Let $F(\cdot)$ be the output of a network. Let $\mathcal{L}(\cdot)$ be the loss function. Let $R$ represent the resource constraints (e.g., model size limitation or latency requirements)

**One-shot NAS.** One-shot NAS techniques [5, 13, 30] proposed to reduce NAS cost by using weight-sharing for all networks in the search space $\mathcal{A}$. One-shot NAS has two important concepts: the "supernet" and the "subnet". The "supernet" refers to a dynamic neural network with dynamic architecture. It encompasses all possible networks in $\mathcal{A}$. The "subnet" refers to a specialized sub-network from the supernet with inherited parameter weights. Each subnet is a part of the supernet and shares network weights with other subnets. Updating the parameters of one subnet affects all other subnets synchronously. Let $W$ and $W_{\alpha_i}$ be the network weights of the supernet and its subnets, respectively. Updating $W_{\alpha_i}$ is equal to updating part of $W$.

One-shot NAS has two optimisation stages: 1) the supernet training stage; 2) the subnet searching stage. The supernet training stage aims to minimize the loss of every subnet via a one-time training:

$$\min_W \sum_{i=0}^{N} \mathcal{L}(F(W_{\alpha_i}; D^{trn})) \tag{1}$$

where $F(W_{\alpha_i}; D^{trn})$ is the output of the subnets $\alpha_i$. Equation (1) can be regarded as a multi-model optimization process, and the bigger the $N$ is, the harder $W$ is to optimise. Given a batch of data, it is really difficult to calculate the losses of all $N$ candidates simultaneously. Thus, SPOS [13] proposed a training strategy that approximates Equation (1) by randomly optimizing $n$ ($n \ll N$) subnets for each mini-batch example $(x, y)$ as Equation (2). As the total number of training iterations spans thousands or even millions, a large scale of subnets will be sampled, trained and aggregated gradients for updating the supernet.

$$\min_W \mathbb{E}_{\alpha_i \in \mathcal{A}} \left[ \sum_{i=0}^{n} \mathcal{L}(F(W_{\alpha_i}; (x, y))) \right] \tag{2}$$

The second stage aims to extract the optimal candidate $\alpha^*$ under given constraints from the well-trained supernet. This process can be formulated as below:

$$\alpha^* = \arg \max_{\alpha \in \mathcal{A}} \text{ACC}(W_\alpha, R; \mathcal{D}^{val}) \tag{3}$$

where $\text{ACC}(\cdot)$ refers to the validation accuracy on $\mathcal{D}^{val}$. To reduce the evaluation cost, recent methods [5, 30, 37, 40] tend to train an accuracy predictor by evaluating a small set of subnets.

One-shot NAS provides a low-cost scheme for training and searching lightweight networks for mobile deployment. However,

compared to training a single DNN, jointly optimising a family of networks is a more data-hungry task, acquiring large amounts of labelled data for supernet training. As for the subnet searching stage, there is also a lack of labelled data for network evaluation or predictor training in sparse-label data contexts.

**Zero-shot NAS.** Zero-shot NAS techniques [7, 21, 25] are an extension of the NAS paradigm that goes a step further by not requiring any parameter training during the architecture search process. These methods rank different networks by designing a specific metric to evaluate or score network architectures. A typical Zero-shot NAS process is as follows:

$$\alpha^* = \arg \max_{\alpha \in \mathcal{A}} \text{Score}(\alpha, R) \tag{4}$$

where $\text{Score}(\cdot)$ represents the scoring function. Different algorithms have different scoring schemes. For example, Zen-NAS [21] scoring network architecture by computing their Gaussian complexity.

Zero-shot NAS provides a non-training strategy for designing a family of lightweight network architectures, which can benefit multi-platform deployment in mobile intelligence. However, training all networks in the family separately before porting is still a resource-expensive task. We note that replacing the data-driven searching steps (i.e., Equation (3)) with zero-shot NAS (i.e., Equation (4)) is an alternative choice in mobile porting.

## 2.2 Semi-supervised Learning

Labelling data is a significant challenge in many real-world scenarios, which often require lots of human labour and expertise knowledge, leading to a situation where the amount of unlabelled examples far exceeds the number of labelled examples. Semi-supervised learning (SSL) offers an effective approach to fully utilize both labelled and unlabelled examples. FixMatch [33] is one of the high-performing and cost-efficient SSL methods in classification tasks, combining consistency regularization [1] and pseudo-labeling [24].

Let $\mathcal{U} = \{u\}$ be the unlabelled training datasets where $u$ is a batch of unlabelled training examples. The loss function of Fix-Match consists of two cross-entropy loss terms: a supervised loss $\mathcal{L}^l(x, y)$ applied to labelled data and an unsupervised loss $\mathcal{L}^u(u)$ for unlabelled data. The supervised loss $\mathcal{L}^l(x, y)$ is a standard loss of labelled examples with weak data augmentation $G^w(\cdot)$:

$$\mathcal{L}^l(x, y) = \mathcal{L}(F(G^w(x)), y) \tag{5}$$

As for the unsupervised loss $\mathcal{L}^u(u)$, FixMatch hypothesizes that the output of weakly-augmented and strongly-augmented unlabelled data should be close. Therefore, FixMatch first calculates the network outputs of unlabelled data with weak augmentation $F(G^w(u))$. Then, it converts those outputs to the probability of predicted classes by the softmax function as pseudo-labels. The $\mathcal{L}^u(u)$ are calculated by pseudo-labels and unlabelled data after strong augmentation $G^s(\cdot)$. Meanwhile, Fixmatch restricts $\mathcal{L}^u(u)$ by setting a minimal confidence threshold $\tau$. The unsupervised loss is valid only if the maximum class probability of a single output in a batch is greater than $\tau$. Therefore, the unsupervised loss $\mathcal{L}^u(u)$ can be formulated as follows:

$$\mathcal{L}^u(u) = \mathbb{I}_\tau \left[ \mathcal{L}(F(G^s(u)), \rho(F(G^w(u)))) \right] \tag{6}$$

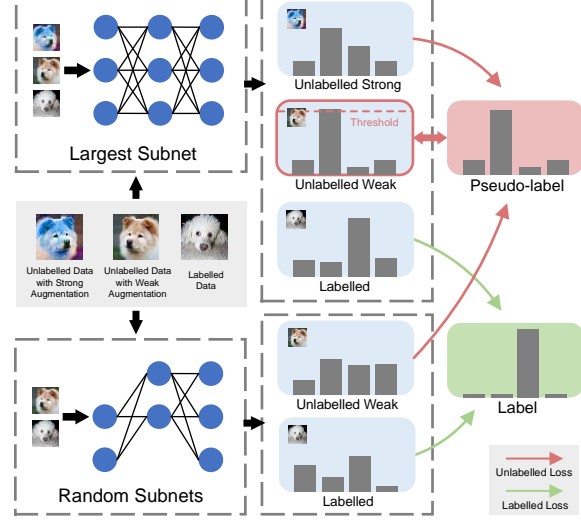

**Figure 3: The semi-supervised-NAS training in MatchNAS**

where $\mathbb{I}_\tau = \mathbb{I}(\max(p(F(G^w(u))) \geq \hat{\tau})$ is the indicator function. The results of $\mathbb{I}_\tau$ is one if the maximum probability of predicted classes is greater than $\tau$; otherwise, it is zero. Note that this judgement acts on each example in a batch of input $u$, and $\hat{\tau} = \{\tau\}_{|u|}$ is the vector of $\tau$ with the same size of $u$. $\rho(\cdot)$ represents the pseudo-labeling.

Although promising results of SSL in label-scarce contexts, we note that the bottleneck of applying SSL to lightweight models is the quality of pseudo-label. The limited network parameters and restricted computing capability make mobile-based lightweight DNNs too small to produce high-quality pseudo-labeling for semi-supervised training. Besides, there is also a lack of systematic study for applying SSL to one-shot NAS training.

## 3 METHODOLOGY

### 3.1 Motivation

To jointly address the challenges associated with model fine-tuning and label scarcity in cloud-edge mobile porting, one intuitive idea is to combine NAS and SSL directly. Given a pre-trained network weights $W$ from cloud served, we directly replace the loss function in Equation (2) with Equations (5) and (6):

$$\min_W \mathbb{E}_{\alpha_i \in \mathcal{A}} \left[ \sum_{i=0}^{n} (\mathcal{L}_{\alpha_i}^l + \mathcal{L}_{\alpha_i}^u) \right] \tag{7}$$

where $\mathcal{L}_{\alpha_i}^l$ and $\mathcal{L}_{\alpha_i}^u$ is the labelled loss and unlabelled loss with specific subnet weights $W_{\alpha_i}$.

However, the majority of networks in mobile porting are lightweight, with fewer layers and limited computational capability. A lightweight network may suffer from low network performance by self-producing low-quality pseudo-labels. Due to weight-sharing, the network weights generated by those low-quality pseudo-labels have a negative impact on the optimization of all other subnets and further adversely affect the optimization of the supernet $W$.

## 3.2 MatchNAS Training

To overcome the bottleneck of low-quality pseudo-labels, we propose our semi-supervised-NAS method, namely MatchNAS. Motivated by knowledge distillation strategy [16], which leverages a large teacher network to "teach" small student networks, our core idea is to select the largest subnet to produce better pseudo-labels for other subnets in each training iteration. For simplification, we also use the symbol $A$ to represent the largest subnet and $W_A$ to represent its network weights.

Figure 3 depicts the semi-supervised-NAS training in MatchNAS. There are three types of training examples: labelled example $(x, y)$, unlabelled example $u$ with weak data augmentation $G^w(u)$ and with strong data augmentation $G^S(u)$. Following Equation (2), MatchNAS samples $n$ subnets for each mini-batch example, including the largest subnet $A$ and $n-1$ random subnets $\{\alpha_1, \ldots, \alpha_{n-1}\}$.

The largest subnet $A$ uses all three examples and obtains three outputs. The output of $G^w(u)$ will be converted to a pseudo-label $\rho_A = \rho(F(W_A; G^w(u)))$. Then, we compute the loss of the largest subnet, including labelled loss $\mathcal{L}_A^l$ and the unlabelled loss $\mathcal{L}_A^u$. This process is similar to Equations (5) and (6):

$$\mathcal{L}_A^l(W_A; (x, y)) = \mathcal{L}(F(W_A; (G^w(x)), y)) \tag{8}$$

$$\mathcal{L}_A^u(W_A; u) = \mathbb{I}_\tau \left[ \mathcal{L}(F(W_A; G^s(u)), \rho_A) \right] \tag{9}$$

As for other sampled subnets $\{\alpha_i\} = \{\alpha_1, \ldots, \alpha_{n-1}\}$, the labelled loss $\mathcal{L}_{\alpha_i}^l$ are similar to Equation (8) with specific weights $W_{\alpha_i}$:

$$\mathcal{L}_{\alpha_i}^l(W_{\alpha_i}; (x, y)) = \mathcal{L}(F(W_{\alpha_i}; (G^w(x)), y)) \tag{10}$$

The difference is that the unlabelled loss of a subnet is computed based on $G^w(u)$ and pseudo-labels $\rho_A$ from the largest subnet $A$ with the indicator function $\mathbb{I}_\tau$ and the minimal confidence threshold $\tau$. This computation is formulated as follows:

$$\mathcal{L}_\alpha^u(W_\alpha; u) = \mathbb{I}_\tau \left[ \mathcal{L}(F(W_\alpha; G^w(u)), \rho_A) \right] \tag{11}$$

We combine Equations (8) to (11) to rewrite the supernet optimization Equation (7) into a new form as below:

$$\min_W \mathbb{E}_{\alpha_i \in A} \left[ \mathcal{L}_A^l + \mathcal{L}_A^u + \sum_{i=0}^{n-1} (\mathcal{L}_{\alpha_i}^l + \mathcal{L}_{\alpha_i}^u) \right] \tag{12}$$

Algorithm 1 demonstrates the training process in MatchNAS.

### 3.3 MatchNAS Searching

After supernet training, we obtain a well-trained supernet containing a huge network family. The next step is to search subnets for porting under given resource constraints. In Section 2.1, we mentioned that popular one-shot NAS methods evaluate a set of subnets with labelled data and then train an *accuracypredictor* for network accuracy prediction as Equation (3).

We note that training a predictor is not suitable for subnet search in mobile porting for two reasons. On the one hand, the training of predictor is costly since it acquires evaluation of a set of subnets (e.g., 1000 subnets in [37]) for a single task. The resource overhead increases linearly as the number of tasks grows. On the other hand, network specialization for mobile porting is a label-limited case, while predictor training necessitates a bunch of labelled data. The performance of the accuracy predictor will dramatically drop without sufficient labelled examples for training.

---

**Algorithm 1:** The training process in MatchNAS

---

**Input:** Supernet $\mathcal{A}$; Pre-trained cloud-based weight $W$;
      Labelled data $\mathcal{D} = (x, y)$ and unlabelled data $\mathcal{U} = u$;
      Strong augmentation $G^s$ and weak augmentation
      $G^w$; Pseudo-labeling $\rho(\cdot)$; The number $n$ of sampled
      subnets in each data batch

Initialize $\mathcal{A}$ with $W$

**while** *not convergence* **do**
    Draw a mini-batch of labelled data $(x, y)$ from $\mathcal{D}$
    Process weak augmentation $G^w(x)$
    Draw a mini-batch of unlabelled data $u$ from $\mathcal{U}$
    Process weak and strong augmentation $G^w(u), G^s(u)$
    Sample the largest subnet $A$
    Calculate $\mathcal{L}_A^l$ with $G^w(x)$ and $y$
    Produce pseudo-label $\rho(G^w(u))$
    Calculate $\mathcal{L}_A^u$ with $G^s(u)$ and $\rho(G^w(u))$
    **for** *i in 1, 2, ..., n − 1* **do**
        Randomly sample subnets $\alpha_i^*$ from $\mathcal{A}$
        Calculate $\mathcal{L}_{\alpha_i^*}^l$ with $G^w(x)$ and $y$
        Calculate $\mathcal{L}_{\alpha_i^*}^u$ with $G^w(u)$ and $\rho(G^w(u))$
    **end**
    Aggregate gradients of $A$ and $\{\alpha_1^*, \alpha_2^*, \cdots, \alpha_{n-1}^*\}$
    Update $W$.
**end**

---

**Algorithm 2:** Zero-shot Search

---

**Input:** Search space $\mathcal{A}$; Zero-shot NAS scorer $\mathcal{S}(\cdot)$;
      Maximum sampled size $M$; scoring batch size $m$;
      Resource constraints $R = \{r_1, \ldots, r_M\}$

Create an empty network collection $C$
**for** $r_i$ *in* $R = \{r_1, \cdots, r_M\}$ **do**
    Random sample a set of networks $\{\alpha_1, \cdots, \alpha_m\}$
    Score sampled networks $\{\mathcal{S}(\alpha_1; r_i), \cdots, \mathcal{S}(\alpha_m; r_i)\}$
    Append the best network $\alpha^*$ into $C$.
**end**

---

To address these problems, we leverage techniques in zero-shot NAS to search subnets for mobile porting. As mentioned in Section 2.1, zero-shot NAS designs an architectural-based metric for network performance evaluation without any parameter training. In this case, we use an architectural scorer $\mathcal{S}(\cdot)$ to efficiently evaluate subnets and pick out the best one under given resource constraints. We name this search process "zero-shot search" and formulate this process in Equation (13). Algorithm 2 provides a meta-algorithm of the searching process.

$$\{\alpha^*\} = \left\{\alpha^* \,\middle|\, \arg\max_{\alpha \in \mathcal{A}} \mathcal{S}(\alpha, R)\right\} \tag{13}$$

## 3.4 MatchNAS with Narrower Search Space

Equation (1) indicates that supernet training is a multi-model optimization task, and a larger $N$ will directly increase its difficulty. Recent work [30] noted that, compared to training a huge network family, training a smaller network family can alleviate the

inference among different size subnets. We hypothesise that our semi-supervised-supernet training will also benefit from a smaller search space. We attempt to leverage the zero-shot NAS techniques to automatically narrow $\mathcal{A}$ before supernet training.

Assuming that we obtain a set of resource constraints $R = \{r_1, \ldots, r_M\}$ for $M$ different platforms by accessing their hardware information and $M \ll N$. We leverage the zero-shot scorer $\mathcal{S}(\alpha_i; r_i)$ to estimate a set of network architectures and extract the one with the highest score. By repeating this step, we obtain a set of high-score networks from the family $\mathcal{A}$ before training. And then we can rebuild a smaller family $\mathcal{A}^*$:

$$\mathcal{A}^* = \left\{(\alpha_i^*, r_i) \,\middle|\, \arg\max_{\alpha \in \mathcal{A}} \mathcal{S}(\alpha_i, r_i)\right\}_M \tag{14}$$

Within such a smaller family $\mathcal{A}^*$, the supernet training can be formulated as a variant of Equation (12):

$$\min_W \mathbb{E}_{\alpha_i \in \mathcal{A}^*}\left[\mathcal{L}_A^l + \mathcal{L}_A^u + \sum_{i=0}^{n-1}(\mathcal{L}_{\alpha_i}^l + \mathcal{L}_{\alpha_i}^u)\right] \tag{15}$$

# 4 EXPERIMENTS

## 4.1 Settings

**Benchmark Datasets** We evaluate the efficacy of MatchNAS on several image classification benchmarks with limited labelled examples to approximate the sparse-label contexts in real-world, including Cifar-10 [19], Cifar-100 [19], Cub-200 [36] and Stanford-Car [18]. Table 1 reports details for these datasets, where "*Train*" indicates the number of training examples; "*Labelled*" indicates the number of labelled training examples; "*Class*" is the number of classes; "*Resolution*" is the image resolution of training examples. The cloud-based network is trained on ImageNet [9] with all labelled data and then transferred to other datasets. We consider 400 labelled examples per class in Cifar-10 and Cifar-100, while 10 labelled examples per class in fine-grained dataset Cub-200 and Stanford-Cars. Those labelled examples will be used to compute labelled loss in Equations (8) and (10), and all training examples will be used to compute unlabelled loss in Equations (9) and (11).

**Table 1: Experimental Datasets**

| Dataset | Train | Labelled | Class | Resolution |
|---|---|---|---|---|
| ImageNet | 1200000 | 1200000 | 1000 | 224 |
| Cifar-10 | 50000 | 4000 | 10 | 32 |
| Cifar-100 | 50000 | 4000 | 100 | 32 |
| Cub-200 | 5994 | 2000 | 200 | 224 |
| Stanford-Cars | 8144 | 1960 | 196 | 224 |

**Search Space** We closely follow the MobileNetV3-Large search space [5, 17] with dynamic macro-structures. We provide dynamic choice for depth $D = \{2, 3, 4\}$, width $W = \{0.5\times, 1.0\times\}$, width expand ratio $E = \{3, 4, 6\}$ and kernel size $K = \{3, 5, 7\}$. The total number of candidate networks is $4 \times 10^{19}$. For more details of this space, please refer to Appendix A.1. Meanwhile, we also generalise MatchNAS to a lighter and more challenging search space Appendix A.2 and experimental results in Table 6.

**Table 2: Comparison of Network Performance in Four Label-limited Data Domains**

| Datasets | Method | SSL | Supernet | Training Cost (GPU Hours) | $\mu$ | Top-1 Accuracy (%) | | |
|---|---|---|---|---|---|---|---|---|
| | | | | | | Smallest | Medium | Largest |
| | OFA | ✗ | ✗ | $500+0.28\times N$ | - | 74.8 | 84.1 | 92.1 |
| | FixMatch | ✓ | ✗ | $1.2\times N$ | 10 | 74.1 | 85.7 | 95.7 |
| Cifar-10 | SPOS | ✗ | ✓ | 0.7 | - | 64.3 | 72.1 | 88.0 |
| | SPOS+FixMatch | ✓ | ✓ | 3 | 10 | 78.9 | 86.8 | 90.4 |
| | **MatchNAS** | ✓ | ✓ | 3 | 10 | **85.8** | **90.2** | **96.5** |
| | OFA | ✗ | ✗ | $500+0.28\times N$ | - | 36.4 | 50.2 | 69.6 |
| | FixMatch | ✓ | ✗ | $1.2\times N$ | 10 | 32.5 | 60.3 | 74.5 |
| Cifar-100 | SPOS | ✗ | ✓ | 0.7 | - | 34.1 | 50.0 | 61.0 |
| | SPOS+FixMatch | ✓ | ✓ | 3 | 10 | 46.6 | 62.1 | 64.9 |
| | **MatchNAS** | ✓ | ✓ | 3 | 10 | **57.9** | **69.8** | **74.9** |
| | OFA | ✗ | ✗ | $500+0.18\times N$ | - | 36.7 | 55.2 | 66.7 |
| | FixMatch | ✓ | ✗ | $0.8\times N$ | 2 | 39.9 | 48.9 | **71.2** |
| Cub-200 | SPOS | ✗ | ✓ | 0.5 | - | 34.2 | 47.8 | 58.6 |
| | SPOS+FixMatch | ✓ | ✓ | 1.2 | 2 | 44 | 54.9 | 62.3 |
| | **MatchNAS** | ✓ | ✓ | 1.2 | 2 | **51** | **61.3** | 70.2 |
| | OFA | ✗ | ✗ | $500+0.18\times N$ | - | 42.9 | 74.1 | 84.7 |
| | FixMatch | ✓ | ✗ | $0.9\times N$ | 4 | 52.5 | 75.9 | 82.2 |
| Stanford-Cars | SPOS | ✗ | ✓ | 0.5 | - | 50.4 | 68.7 | 78.4 |
| | SPOS+FixMatch | ✓ | ✓ | 1.4 | 4 | 53.6 | 71.1 | 77.9 |
| | **MatchNAS** | ✓ | ✓ | 1.4 | 4 | **60.4** | **78.9** | **86.8** |

**Mobile Platforms** For on-device evaluation, we prepare four high-performing smartphones via Samsung Remote Test Lab [31], including Samsung Galaxy S23, Galaxy S22, Galaxy Note 20 and Galaxy A12, as shown in Table 3. Their computing ability decreases in the order in which they are listed. We measure the actual latency for each model with a batch size of 1 using the Pytorch-mobile framework [28] on the Android 13 operating system. An example of the on-device test can be found in Appendix B. The results of the on-device evaluation are shown in Section 4.5.

**Table 3: Hardware Platforms**

| Platforms | SoC | RAM | Year |
|---|---|---|---|
| Samsung Galaxy S23 | Snapdragon 8 Gen 2 | 8GB | 2023 |
| Samsung Galaxy S22 | Exynos 2200 | 8GB | 2022 |
| Samsung Galaxy Note 20 | Exynos 990 | 8GB | 2020 |
| Samsung Galaxy A12 | Mediatek Helio P35 | 3GB | 2020 |

## 4.2 Training Details

The core idea of MatchNAS is the combination of semi-supervised training and one-shot NAS training. In practice, we combine the pseudo-labelling technique in FixMatch [33] and the supernet training skill in SPOS [13] to produce our semi-supervised-NAS training scheme. In the next section, we provide detailed comparisons of MatchNAS and these two methods.

We note that the loss term of semi-supervised training is applicable for one-shot NAS since it has a low computing cost and fully utilizes unlabelled examples. The training process of Match-NAS can be simplified as a transfer task, including three steps: 1) train a cloud-based network, 2) semi-supervised-NAS training for label-scarce datasets and 3) search subnets under different resource constraints.

We first train a cloud-based network on ImageNet for 180 epochs with a learning rate of 8e-2. This network is a full network in search space with the maximum architecture $\{D = 4, W = 1.0\times, E = 6, K = 7\}$ in each stage. The training duration is about 150 GPU hours measured on an NVIDIA RTX 3090. Then, we transform the cloud-based model to a supernet with varying architectural configurations as Table 5 and fine-tune this supernet to different datasets with different data domains and limited data labels.

As for the semi-supervised-NAS training, we train a supernet for 50 epochs, using Adam optimizer with weight decay $3 \times 10^{-5}$. The initial learning rate is $3 \times 10^{-4}$ with a cosine schedule for learning rate decay [22]. The training batch size is $16 + 16 \times \mu$, where 16 is the batch size of labelled data and $\mu$ is the ratio of labelled data and unlabelled data. For each iteration, the training process is as Algorithm 1, and we set $n = 4$. The pseudo-label's confidence threshold ($\tau$) is 0.95, and we provide an ablation study of this setting in Section 5.2. The weak and strong data augmentation process follows the settings in FixMatch [33].

## 4.3 Network Performance

In this section, we compare MatchNAS with art DNNs baselines on four different datasets. We consider the following popular networks or network combinations as baselines: (a) Cloud-trained one-shot NAS method OFA [5], which firstly trains a supernet on ImageNet and sample subnets for further fine-tuning with labelled data; (b) Semi-supervised method FixMatch [33] using both labelled and unlabelled data; (c) Transfer-trained one-shot NAS method SPOS [13] train a supernet using labelled data; (d) SPOS+FixMatch a directed

combination of NAS and SSL as Equation (7) using both labelled and unlabelled data. For a fair comparison, all methods inherit weights from the same cloud-based network, and each method uses the same MobileNetV3-Large search space as mentioned in Section 4.1.

We summarize experimental results in Table 2. "SSL" indicates whether a method uses both labelled and un-labelled data or only labelled data. "Supernet" indicates whether a method trains a supernet or a single DNN. "Training Cost" represents the training duration measured by an NVIDIA RTX 3090 GPU. "$N$" is the number of possible deployment platforms. Compared to other methods, OFA first trains a supernet on ImageNet, resulting in an extra 500 GPU hours time cost. For non-supernet methods, the total training cost is calculated by the average cost of training a single DNN times $N$. "$\mu$" represents the ratio of labelled data and unlabelled data in one training iteration. We set different $\mu$ for different datasets towards their different ratios of labelled data (see Table 1).

We observe the Top-1 accuracy for three different sizes of networks trained by different methods. The "largest" is the largest subnet in the search space with architecture configuration $\{D = 4, W = 1.0\times, E = 6, K = 7\}$, while the "smallest" is the smallest subnet $\{D = 2, W = 0.5\times, E = 3, K = 3\}$. The configurations of the "medium" are $\{D = 4, W = 0.5\times, E = 6, K = 5\}$. For methods using supernet training, these networks inherited weights from the supernet directly. For non-supernet methods, these networks will be trained from scratch separately.

We can see that MatchNAS significantly outperforms all baselines in the smallest and medium subnet on all four datasets, with about a minimum of 4% and a maximum of 20% accuracy improvement. These experimental results prove the effectiveness of Match-NAS in porting lightweight models with label-scarce datasets. As for the largest network, MatchNAS report a competitive network performance compared to FixMatch. Note that MatchNAS provides a supernet with numerous subnets, while FixMatch need repeated network training for different platforms, which is resource-consuming.

As for the training cost, MatchNAS training a supernet containing $4\times10^{19}$ candidate subnet via a one-time training. As the number $N$ of possible deployment platforms increases, MatchNAS can save significant time overhead than non-supernet methods. Compared to the supernet in SPOS, MatchNAS utilize the SSL techniques to improve network performance dramatically.

In summary, MatchNAS provide a better trade-off between the training cost and the network performance in sparse-label contexts.

## 4.4   Experiments with Varying Labelled Data

In this section, we perform experiments on Cifar10 with extremely limited labelled data in MobileNetV3-Large search space. Except for 4000 labelled data in Table 2, we further consider 250 and 50 labelled data, i.e., 25 and 5 labelled data per class. Obviously, the less labelled data the dataset contains, the more difficult the training is.

Table 4 reports the Top-1 accuracy results of networks trained in five different methods. "smallest" and "largest" represent the smallest subnet and the biggest subnet in the search space. Match-NAS reports the highest accuracy 95.8%, 95.1%, 83.6% of the largest subnet and 85.8%, 86.9%, 70.6% of the smallest subnet in three label-scarce settings. These experimental results further justified the effectiveness of MatchNAS.

**Table 4: Performance Comparison in Cifar10 with Different Numbers of Labelled Examples**

| Model | Smallest Top-1(%) | | | Largest Top-1 (%) | | |
|---|---|---|---|---|---|---|
| | 4000 | 250 | 50 | 4000 | 250 | 50 |
| OFA | 74.8 | 44.4 | 11.0 | 92.1 | 76.7 | 50.8 |
| FixMatch | 74.1 | 60.0 | 31.9 | 95.7 | 95.1 | 81.7 |
| SPOS | 64.3 | 35.7 | 16.6 | 88.0 | 66.9 | 37.9 |
| SPOS+FixMatch | 78.9 | 72.7 | 44.1 | 90.4 | 90.3 | 65.1 |
| MatchNAS | **85.8** | **86.9** | **70.6** | **96.5** | **95.1** | **83.6** |

## 4.5   On Device Performance

After supernet training, we carry out a zero-shot search to sample high-performing subnets for mobile deployment as Algorithm 2. We closely follow Zen-NAS [21] to compute the Gaussian complexity for scoring subnets. For each resource constraint setting (e.g., FLOPs limits), we randomly sample 20 subnets for evaluation and select the best one with the highest Gaussian complexity score. The search cost for each subnet is less than one GPU minute.

We compare MatchNAS's subnets with other subnets using different training strategies, including FixMatch and SPOS+FixMatch. Subnets from MatchNAS and SPOS+FixMatch are sampled from their own supernet, and networks from FixMatch are trained separately. For a fair comparison, networks from different methods have similar FLOPs constraints.

Figure 4 reports the performance of latency-accuracy trade-off on four datasets and devices. MatchNAS consistently achieve comparable and higher network performance and produces a better accuracy-latency trade-off by training a single supernet with limited labelled data. Compared to SPOS+FixMatch, which also trains a supernet, MatchNAS reports a superior subnet performance. Notice that the network training cost of MatchNAS is much lower than FixMatch as the number of platforms increases. Compared to FixMatch, MatchNAS shows a better accuracy-latency trade-off among low-latency networks while competitive performance among high-latency networks.

## 4.6   Experiments with a Narrower Search Space

In Section 3.4, we propose a search space narrowing method as Equation (15). The core idea is to select a set of high-quality subnets as a narrower space before supernet training to reduce the difficulty of optimization. In this section, we report a subnet performance comparison between using a narrower search space and not.

In practice, we sample 200 lightweight subnets ranging from 50M to 90M FLOPs based on Gaussian complexity [21] and build a narrower and smaller search space. Each selected subnet is selected with the highest Gaussian complexity score from twenty random subnets. Figure 5 (a) reports a performance comparison of those 200 subnets from three different supernets, and MatchNAS[†] represents the supernet training within the narrower search space. Clearly, MatchNAS[†] reports about 2% higher accuracy compared to Match-NAS. These phenomena verify our hypothesis in Section 3.4 that a narrower search space can be optimized more easily than a large one while the number of available subnets decreases.

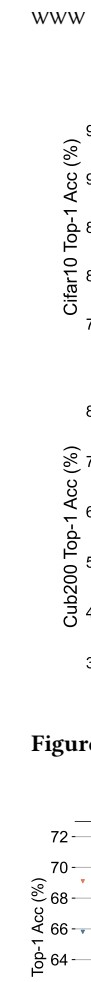
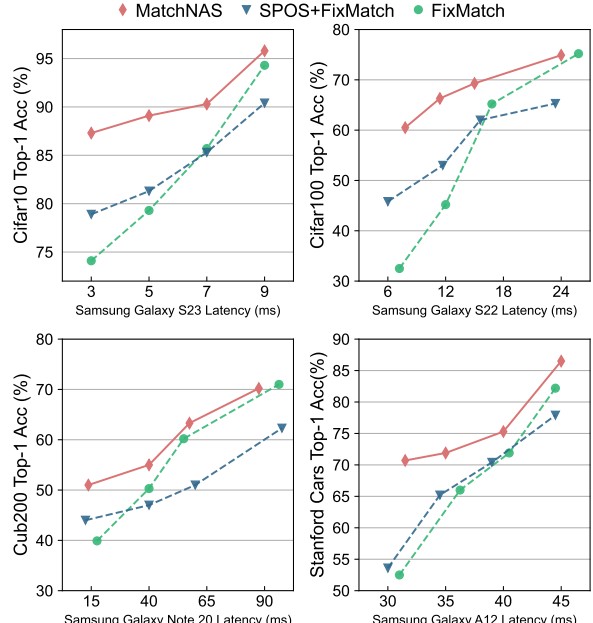

Figure 4: Latency-accuracy trade-off on four mobile devices.

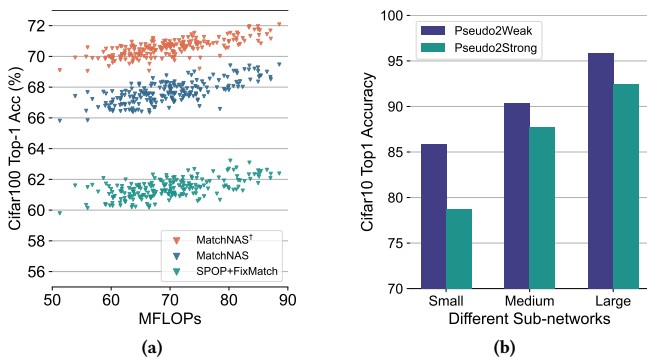

(a)      (b)

Figure 5: (a): Network performance of a set of subnets on Cifar-100; (b): Performance comparisons of different unlabelled loss types.

## 5 ABLATION STUDY

### 5.1 Unlabelled Loss

In Section 3.2 and Figure 3, we mentioned that the unlabelled loss of subnets is computed by the pseudo-label and unlabelled example with weak augmentation as Equation (15). In this section, we replace the weak augmentation with the strong one and Figure 5 (b) shows the results. We compare three subnets with different model sizes from MatchNAS. "Pseudo2Weak" represents the supernet's unlabelled loss computed by the pseudo-label and unlabelled example with weak augmentation, while "Pseudo2Strong" is with strong augmentation. The "Pseudo2Weak" reports a higher subnet performance, especially in small subnets. It also indicates that lightweight networks have difficulty in handling complex inputs.

## 5.2 Confidence Threshold

The confidence threshold $\tau$ controls the trade-off between the quality and quantity of pseudo-label in the loss term Equations (9) and (11). The output prediction can be converted to a pseudo-label when the model assigns a probability to any class that is above the threshold. The value of $\tau$ ranges from 0 to 1, where $\tau = 0$ means all predictions are pseudo-label, and a larger $\tau$ means predictions with higher class confidence can be converted to pseudo-label.

Previous work [20, 33] has proved that the quality of pseudo-label contributes more to the network performance than the quantity. We validate the effectiveness of the confidence threshold in our SSL-based supernet training, and Figure 6 (a) show report a comparison of subnet performance with different values of $\tau$. $\tau = 0.95$ report a accuracy improvement of about 1.5% compared to $\tau = 0.0$.

### 5.3 One-shot NAS Techniques

MatchNAS is a combination of SSL and one-shot NAS techniques. In this section, we alternate SPOS with another supernet training technique in BigNAS [41]. Given a minibatch of data, BigNAS optimise both the largest and the smallest subnet in the search space and $n - 2$ random-sampled subnets. Figure 6 (b) depicts subnet performance from three different methods. "MatchNAS-BigNAS" is the combination of MatchNAS and BigNAS, which show higher performance in low FLOPs compared to the vanilla MatchNAS. This phenomenon is mainly caused by the optimization strategy in BigNAS. The experiment results indicate that MatchNAS can alternatively combine with other NAS methods except methods demanding lots of labelled data [5, 37].

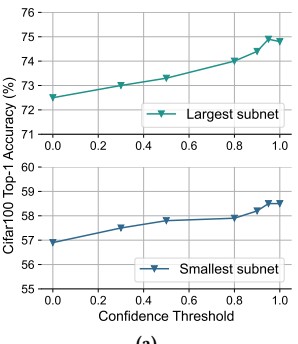
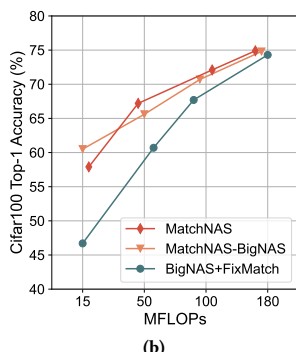

(a)      (b)

Figure 6: (a): Network performance comparisons of using different thresholds; (b): Subnets performance comparisons of three different methods under different FLOPs constraints.

## 6 CONCLUSIONS

In this paper, we provide MatchNAS to optimize edge AI by automating porting lightweight mobile networks in sparse-label data contexts. Our algorithm leverages NAS and SSL techniques to optimise mobile porting in spare-label data contexts. We demonstrate the effectiveness of MatchNAS for mobile deployment in different image classification tasks with promising experimental results. We hope our approach will inspire more researchers toward a deeper understanding of DNNs for edge AI.

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

# A SEARCH SPACE

## A.1 The Vanilla Search Space

We closely follow the MobileNetV3-Large Search Space [5, 17]. Our search space is shown in Table 5. Except for the fixed architecture head and tail, there are five repeated macro-structures with dynamic configurations named DMBConv, which refer to the inverted dynamic residual block. *Depth* represents the number of dynamic convolution blocks (or layers) in the dynamic stage. *Width* and *Expand* denote the output channel width of each block and the width expanding ratio. The maximum channel width is calculated by *Width × Expand*. *Kernel* is the kernel size of each block. To further reduce the computation complexity to meet the demands of lightweight mobile deployment, we further provide two sets of width choices, including {12, 20, 40, 56, 80} (0.5×) and {24, 40, 80, 112, 160} (1.0×). The total number of candidate subnets is $((3 \times 3)^2 + (3 \times 3)^3 + (3 \times 3)^4)^5 \times 2 \approx 4 \times 10^{19}$. As different input resolutions, the computing complexity of the largest and the smallest subnet in Cifar-10 and Cifar-100 (32 × 32) are about 180M FLOPs and 15M FLOPs, while in Cub-200 and Stanford-Cars (224 × 224), they are 560M and 58M FLOPs.

Table 5: MobileNetV3-Large Search Space with dynamic network configurations.

| Stage | Depth | Width | Expand | Kernel |
|---|---|---|---|---|
| Conv | 1 | 16 | - | 3 |
| MBConv | 1 | 16 | 1 | 3 |
| DMBConv1 | {2, 3, 4} | {12, 24} | {3, 4, 6} | {3, 5, 7} |
| DMBConv2 | {2, 3, 4} | {20, 40} | {3, 4, 6} | {3, 5, 7} |
| DMBConv3 | {2, 3, 4} | {40, 80} | {3, 4, 6} | {3, 5, 7} |
| DMBConv4 | {2, 3, 4} | {56, 112} | {3, 4, 6} | {3, 5, 7} |
| DMBConv5 | {2, 3, 4} | {80, 160} | {3, 4, 6} | {3, 5, 7} |
| Conv | 1 | 960 | - | 1 |
| Conv | 1 | 1280 | - | 1 |

## A.2 Generalising MatchNAS to Other Space

MatchNAS, which focus on edge AI with limited computing resource, is expected to apply to more lightweight architectures. Except for the vanilla search space in Appendix A.1 ranging from 15M to 180M FLOPs, we further design a smaller search space ranging from 4M to 75M FLOPs. As shown in Table 6, this search space is based on MobileNetV3-Small, containing only four dynamic stages with a narrower network width and more dynamic width choices, including {12, 20, 40, 56, 80} (0.5×), {12, 20, 40, 56, 80} (1.0×) and {36, 60, 72, 144} (1.5×).

We generalise MatchNAS to this more compact search space, and other training settings are similar. Table 7 report a performance comparison in Cifar-10 with 4000 labelled example. The "largest" is the largest subnet in the search space with architecture configuration $\{D = 4, W = 1.5\times, E = 6, K = 5\}$, while the "smallest" is the smallest subnet $\{D = 2, W = 0.5\times, E = 3, K = 3\}$. The configurations of the medium one, "medium", are $\{D = 4, W = 1.0\times, E = 6, K = 5\}$.

As a supernet, MatchNAS provides about $1 \times 10^{13}$ different candidate subnets after one-time training and reports a competitive network performance. In the most extremely lightweight case, MatchNAS report a 8% higher accuracy compared to baseline supernet SPOS+FixMatch and 10% higher accuracy compared to standard DNN FixMatch.

Table 6: MobileNetV3-Small Search Space with dynamic network configurations.

| Stage | Depth | Width | Expand | Kernel |
|---|---|---|---|---|
| Conv | 1 | 16 | - | 3 |
| MBConv | 1 | 16 | 1 | 3 |
| DMBConv1 | {2, 3, 4} | {12, 24, 36} | {3, 4, 6} | {3, 5} |
| DMBConv2 | {2, 3, 4} | {20, 40, 60} | {3, 4, 6} | {3, 5} |
| DMBConv3 | {2, 3, 4} | {24, 48, 72} | {3, 4, 6} | {3, 5} |
| DMBConv4 | {2, 3, 4} | {48, 96, 144} | {3, 4, 6} | {3, 5} |
| Conv | 1 | 576 | - | 1 |
| Conv | 1 | 1024 | - | 1 |

Table 7: Performance Comparison of Top-1 Acc

| Model | SSL | Supernet | Cifar-10 Top-1 Accuracy(%) | | |
|---|---|---|---|---|---|
| | | | Smallest | Medium | Largest |
| FixMatch | ✓ | ✗ | 62.6 | 85.8 | 94.3 |
| SPOS | ✗ | ✓ | 53.6 | 76.5 | 83.4 |
| SPOS+FixMatch | ✓ | ✓ | 66.2 | 82.9 | 86.3 |
| MatchNAS | ✓ | ✓ | **72.3** | **89.1** | 93.9 |

# B EXAMPLE ON-DEVICE EVALUATION

Figure 7 shows an example evaluation on smartphones. All on-device tests are performed on the Samsung Remote Test Lab [31].

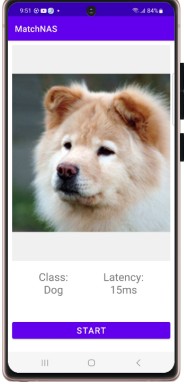

Figure 7: Example evaluation on Samsung Galaxy Note 20. This result is produced by Samsung Remote Test Lab.

