# OpenReview forum: "MatchNAS: Optimizing Edge AI in Sparse-Label Data Contexts via Automating Deep Neural Network Porting for Mobile Deployment"
_ACM.org/TheWebConf/2024/Conference — TheWebConf24_

### Official Review · Reviewer_HPf6 · 2023-11-09

**Novelty:** 6
**Technical Quality:** 6

**Review:**

This paper presents MatchNAS, a neural network porting scheme designed to automate the process for mobile platforms, especially in contexts with scarce labeled data.

Pros\
S1: The significance of this work lies in its potential to bridge the gap between cloud AI and edge AI, improving the efficiency of porting networks to edge devices. This is particularly relevant as edge computing grows with the expansion of IoT and mobile devices.\
S2: The paper introduces MatchNAS, which appears to be a novel approach combining NAS schemes and semi-supervised learning techniques to reduce the effort in network fine-tuning and address the challenge of limited labeled data. \
S3: The paper provides a clear methodology, making it replicable for further research and practical application.

Cons\
W1: The paper may lack detailed discussion on limitations and potential trade-offs of the MatchNAS approach.

**Questions:**

Q1: What is the limitation of the proposed MatchNAS model?

**Reviewer Confidence:**

2: The reviewer is willing to defend the evaluation, but it is likely that the reviewer did not understand parts of the paper

**Scope:**

4: The work is relevant to the Web and to the track, and is of broad interest to the community

---

### Official Review · Reviewer_CPPQ · 2023-11-11

**Novelty:** 6
**Technical Quality:** 6

**Review:**

This paper proposes to combine neural architecture search techniques with mobile deployment of DNNs and proposes a new network porting scheme MatchNAS. MatchNAS transforms a trained DNN to a supernet and conducts a semi-supervised NAS training process to transfer the supernet to a label-scarce dataset. Experimental results shows both the effectiveness and the training efficiency of MatchNAS. The on device performance further validate its network performance.

The paper is generally well structured and there are comprehensive experiments on the proposed MatchNAS. Since not being an expert in either edge AI nor NAS field, this reviewer couldn't discern any obvious weaknesses within this paper.

**Questions:**

Since the authors have mentioned the efficiency of the proposed MatchNAS, it would be more persuasive to explicitly analyze the computational complexity of MatchNAS.

**Reviewer Confidence:**

1: The reviewer's evaluation is an educated guess

**Scope:**

4: The work is relevant to the Web and to the track, and is of broad interest to the community

---

### Official Review · Reviewer_1ciL · 2023-11-17

**Novelty:** 3
**Technical Quality:** 4

**Review:**

In essence, the paper explores the deployment of large models trained on the cloud to edge devices. It acknowledges the challenges posed by the diversity and resource constraints of various edge devices. To address this, the paper introduces a novel application of Neural Architecture Search (NAS), term MatchNAS.

Strengths
+ The paper addresses an important practical problem in the field of ML deployemnt.
+ Overall, the paper is easy to follow.

Weaknesses
- The novelty of the work could be further emphasized. While the application of teacher training to NAS is insightful, it is widely employed in zero-shot and few-shot NAS algorithms. This aspect makes the paper seem like a deployment specificaiton with application of existing techniques.
- A comparison with more recent works that utilize self or semi-supervised learning to mitigate the data scarcity problem in NAS would have been beneficial like [1,2].
- The inclusion of additional benchmarks datasets considered by the NAS community, such as Pascal VOC and Cityscapes, would have enriched the evaluation process

[1]Semi-Supervised Neural Architecture Search. Renqian Luo, Xu Tan, Rui Wang, Tao Qin, Enhong Chen, and Tie-Yan Liu. CoRR abs/2002.10389 (2020).
[2]Self Semi Supervised Neural Architecture Search for Semantic Segmentation. Loïc Pauletto, Massih-Reza Amini, and Nicolas Winckler. CoRR abs/2201.12646 (2022).

**Questions:**

Please see the weaknesses pointed out in the Review

**Reviewer Confidence:**

3: The reviewer is confident but not certain that the evaluation is correct

**Scope:**

4: The work is relevant to the Web and to the track, and is of broad interest to the community

---

### Official Review · Reviewer_oTat · 2023-11-24

**Novelty:** 3
**Technical Quality:** 3

**Review:**

This paper presents a semi-supervised-network-architecture-search algorithm, named MatchNAS, which combines NAS and SSL methods to automate network porting for mobile platforms in label-scarce contexts.

Strength
* The motivation is relatively clear and reasonable.
* Paper is well-written.


Weakness

* The main concern is novelty. It seems that it only combines NAS and SSL technology, lacking technological innovation points.

* The experiment was only based on the MobileNetV3 structure, and it is unclear whether this method is applicable to networks with other structures.

* Abbreviated words need to be marked with their full name where they first appear, e.g., SSL in Introduction.

**Questions:**

Please see weakness.

**Reviewer Confidence:**

3: The reviewer is confident but not certain that the evaluation is correct

**Scope:**

4: The work is relevant to the Web and to the track, and is of broad interest to the community

---

### Official Review · Reviewer_qmLc · 2023-12-01

**Novelty:** 3
**Technical Quality:** 4

**Review:**

This paper proposes an approach for Deep network porting in mobile devices when the enough labelled data is not available.

The authors essentially combines semi-supervised learning and neural architecture search to achieve the goal.

The paper has limited novelty. All one has to do apply semi-supervised learning and then compress the network to fit in mobile platform.

I did not find any rationale behind the use of architecture search student-teacher paradigm to achieve this. NASes are already unstable and in combination with small labeled data this makes more unreliable.

Rather a straightforward approach would be to train a network in semi-supervised settings and prune it within the fixed budget for deployment.

**Questions:**

Did the authors consider pruning based network fine-tuning?

What was the performance of the NAS model used by the authors on data with varying labels?

What happens if we simpler approach like this: train a Semi-supervised model and then use network pruning to compress the network?

**Ethics Review Description:**

None.

**Reviewer Confidence:**

4: The reviewer is certain that the evaluation is correct and very familiar with the relevant literature

**Scope:**

4: The work is relevant to the Web and to the track, and is of broad interest to the community

---

### Official Review · Reviewer_btnQ · 2023-12-01

**Novelty:** 5
**Technical Quality:** 4

**Review:**

This paper presents a neural network porting scheme, named MatchNAS, to automate network porting for mobile platforms. MatchNAS addresses two bottlenecks present in existing techniques: need for large amount of data and difficulty in producing high-quality artificial labels. MatchNAS' evaluation on four image classification datasets show good performance improvement. Further, its deployment show better latency-accuracy trade-off compared to baselines.

Strengths:
- The paper clearly describes technical challenges with existing techniques and explains the novelty of the proposed method w.r.t to these techniques
- Related works on Neural Architecture Search (NAS) explains in detail the preliminaries necessary to understand the main contribution
- Results compared to SOTA are promising

Weaknesses:
- Whereas the paper describes the contributions in context of mobile network porting, the broader applicability of the contribution to the Web is not clear. The technique as described looks generalisable but generalisability aspect needs to be made explicit.
- The proposed approach is about mobile networks but the evaluation is on image classification datasets. Why the four datasets employed in evaluation are relevant is not clarified.

**Questions:**

- What are the unique challenges in the domain of mobile number porting that differentiates it from other domain? <Pg1, right column> does briefly touches upon it but I ask this to better understand the generalisability of the proposed approach.
- Why the four image datasets employed in evaluation are relevant for evaluating technique specific to mobile network porting?

**Ethics Review Description:**

-

**Reviewer Confidence:**

1: The reviewer's evaluation is an educated guess

**Scope:**

3: The work is somewhat relevant to the Web and to the track, and is of narrow interest to a sub-community

---

### Decision · Program_Chairs · 2024-01-22

**Decision:**

Accept

**Comment:**

Summary: A scheme to automate the porting of deep-learning models from cloud servers to resource-constrained edge devices.


 Strengths:
 + Address an important practical challenge
 + New method using NAS and SSL
 + Clear technical motivation
 + Easy to follow


 Weaknesses:
 - Concerns about limited novelty
 - Concerns about generalizability of the methods
 - Rationale for using NAS is less clear
 - Better related work would be helpful
 - Need to discuss limitations


 Recommendation: The paper addresses a useful practical problem and has a method, but some elements of novelty and rationale are debated by the reviewers.